# Potential of Supercritical *Acrocomia aculeata* Oil and Its Technology Trends

**Giselle Cristine Melo Aires** and **Raul Nunes de Carvalho Junior** *

Extraction Laboratory, Post-Graduation Program in Food Science and Technology (PPGCTA), Institute of Technology (ITEC), Federal University of Pará (UFPA), Belém 66075-110, PA, Brazil
* Correspondence: raulncj@ufpa.br; Tel.: +55-91-982041574

**Abstract:** This study discusses the bioactive composition, supercritical extraction, and biological activity of *Acrocomia aculeata* in publications in the last ten years. Numerous compounds have been identified in *A. aculeata*, which include fatty acids, carotenoids, phenolic compounds, and tocopherols, discussed in this article. Although there are several studies with the fruit using conventional extraction to obtain oil, there are only a few involving extraction at high pressures. Therefore, this article emphasized the potential of extraction with supercritical fluid (SFC) to obtain oil due to its good selectivity, fractions distributed in terms of mass yield, and chemical composition of the obtained extracts, which provides a solvent-free product, making it safe for application in the food industry. The biological activity of *A. aculeata* extracts was also discussed, including antidiabetic, anti-inflammatory, diuretic, antimicrobial, antioxidant, neuroprotective, and photoprotective effects, which can produce effects on human health. This review produces important results that can act as a basis for future studies related to obtaining bioactive compounds from *A. aculeata* with a high degree of purity and good quality in its applications.

**Keywords:** *Acrocomia aculeata*; bioactive compounds; supercritical extraction; biological activity





## 1. Introduction

*Acrocomia aculeata (Jacq.) Lodd* is a common palm tree in the Cerrado region, but it can be found throughout the Brazilian territory. The plant, also known as macaúba, macaíba, or bocaiuva, is a species with great economic potential for the production of oil for the chemical, pharmaceutical, cosmetic, energy production, and food industries, in addition to being considered an important source of nutrients [1–4]. The fruit of macaúba has a pulp that is a great source of nutrients, including carbohydrates, fibers, proteins, vitamins, and minerals, used in human and animal nutrition. In addition, the pulp of the macaúba fruit contains bioactive compounds, such as unsaturated fatty acids, tocotrienols, carotenoids, phenolics, and flavonoids, which have antioxidant and anti-inflammatory properties [5–7].

Several studies have demonstrated the health benefits generated by consuming the pulp of macaúba fruit, including reducing the risk of cardiovascular diseases, diabetes, and some types of cancer, as well as improving gastrointestinal and liver function. In addition, consuming the pulp of macaúba fruit can help increase the feeling of satiety and control appetite, which can be beneficial for weight loss and maintaining a healthy lifestyle [8–11].

The biological effects of the macaúba fruit pulp have also been investigated in animal and cell culture studies, with promising results regarding anti-inflammatory, antioxidant, and neuroprotective activity. In addition, the oil extracted from the pulp and endocarp of the macaúba fruit has antimicrobial and anti-inflammatory properties, suggesting its potential use in the pharmaceutical and cosmetic industries [4,11,12].

In order for *Acrocomia aculeata* to fully provide its economic potential in the food and pharmaceutical industries, it is necessary to elucidate extraction methods that produce a pure, higher quality, solvent-free oil on a large scale while preserving its bioactive

characteristics. In that regard, the application of supercritical technology becomes an excellent possibility for obtaining oil; therefore, this study aims to evaluate the nutritional composition, phytochemical composition, and human health effects of *Acrocomia aculeata* fruit oil [13–15].

## 2. Botanical Taxonomy (*Acrocomia aculeata*)

The genus *Acrocomia* can be found in a wide geographic distribution due to its adaptation to different climates [1]. *Acrocomia aculeata (Jacq.) Lodd.* is a palm tree with oily fruits, also known as macaúba, arara, or bocaiuva. It is common in tropical and subtropical regions, and it also can be found throughout the Brazilian territory. This species blooms more intensely from September to November, and its fruits turn mature and ready for harvesting between 12 and 14 months after flowering, when they detach from the bunch and fall on the ground [13]. Macaúba fruits are arranged in clusters and have a spherical shape, with a smooth skin that can vary from brown to yellowish when ripe, as shown in Figure 1. Its pulp is yellow and adheres to the endocarp that encloses the kernel and can be consumed fresh or have its oil extracted. Its kernel also contains high-quality and colorless oil that can be used in the food, cosmetic, and pharmaceutical industries [16].

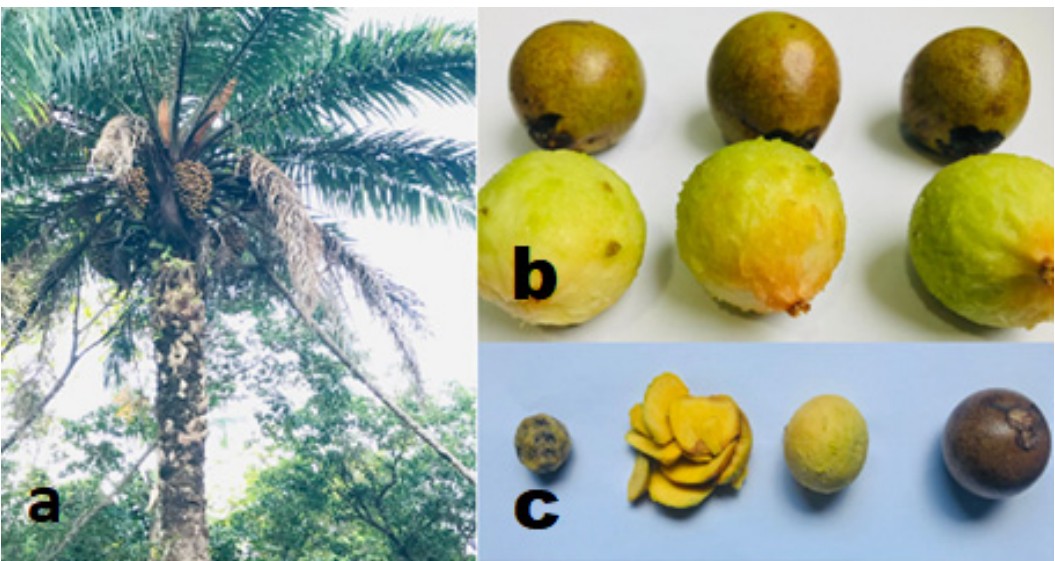

**Figure 1.** *Acrocomia aculeata*: (**a**) palm trees, (**b**) fruit, (**c**) pulp, kernel, and skin.

## 3. Nutritional Composition

*Acrocomia aculeata*, also known as macaúba or bocaiuva, is a common and economically important palm species in Brazil, especially in the state of Mato Grosso do Sul. Several studies have explored its nutritional properties, both of the pulp and kernels, as shown in Table 1, revealing its potential as a source of nutrients and vegetable oils [1]. However, this fruit is still exploited in an extractive manner, and the process of removing the skin is on a small scale [16].

Macaúba pulp is rich in β-carotene, as well as copper, potassium, and zinc, and can be classified as a source of these minerals for children and adults [6]. The macaúba almond oil (*Acrocomia aculeata*) is rich in lipids and proteins [1,17,18]. Bocaiuva almond flour has a high protein (38.0%) and fiber (45.3%) content, even though it is deficient in essential amino acids such as threonine, histidine, and leucine. The protein in bocaiuva almonds has a biological value similar to casein but less capacity to promote growth and digestibility [5]. Macaúba is a fruit with a high energy value and low moisture, mainly due to the high concentration of lipids in both the pulp and the kernel [13,19].

**Table 1.** Proximal composition, energy, and minerals of pulp and kernel of *A. aculeata* [1].

| Composition | Pulp | Kernel |
|---|---|---|
| Protein (g/100 g) | 3.61–8.4 | 16.49–21.88 |
| Lipids | 31.88–43.22 | 52.08–56.67 |
| Carbohidrate (g/100 g) | 8.55–19.75 | 8.54–9.86 |
| Ash | 3.63–4.95 | 1.84–2.11 |
| Moisture (g/100 g) | 35.42–40.97 | 14.24–16.19 |
| Fiber (g/100 g) | - | - |
| **Energy (kJ/100 g)** | 1891.37 | 2522.30 |
| **Minerals** | | |
| K | 75–241.2 | 28.66–60.39 |
| P | 3.84–19.81 | 27.85–92.51 |
| Ca | 8.29–32.85 | 4.82–30.06 |
| Mg | 3.71 153.35 | 6.87–57.2 |
| Zn | 0.3–1.00 | 1.8–9.7 |
| Cu | 1.10–2.3 | 0.8–1.7 |
| Fe | 0.8–5.4 | 1.2–11.9 |
| Mn | 0.4–1.7 | 1.00–5.6 |
| **Total carotenoids (β-carotene) μg/g** | 694 ± 8.31 | 26.1–36.1 |

However, the pulp and almonds of macaúba palms are rich in carotenoids and tocopherols, making them an important source of vitamins A and E [17,20]. In that regard, bocaiuva pulp flour can be used as a supplement in school feeding programs, offering nutrients and vitamins of natural origin [18]. The high fiber content in the studied pulps and almonds also indicates their potential to be used in the formulation of bakery products, enriching their texture, flavor, and nutritional value [17].

## 4. Phytochemical Composition

Phytochemical composition refers to the presence and quantity of bioactive compounds in foods and plants. The palm trees mentioned earlier have various bioactive compounds, including fatty acids, phenolic compounds, carotenoids, and tocopherols.

### 4.1. Fatty Acids

Fatty acids are organic compounds that have a structure formed by a hydrocarbon chain with a carboxyl group (-COOH) at one end; they are classified as saturated or unsaturated. Their intake is essential for the body's functioning, as they provide energy and participate in the structural composition of cell membranes. Their consumption is linked to some biological effects [10]. The oil from the pulp and seeds has a high concentration of fatty acids, as shown in Table 2.

Macaúba pulp is rich in unsaturated fatty acids, mainly oleic acid, which represents more than 50% of the total fatty acids. It acts in the reduction of an important compound that helps to control blood pressure and helps to reduce triglycerides and cholesterol (LDL) without reducing HDL levels in the blood [14,17,18]. *A. aculeata* pulp oil is rich in monounsaturated fatty acids, especially oleic acid (71%), accompanied by a high concentration of palmitic acid; these are the most abundant fatty acids in the mesocarp, as shown in Table 2. In kernel oil, lauric, oleic, myristic, and palmitic acids are prevalent [12,16]. Several studies suggest this fatty acid has potential anti-inflammatory, diuretic, and antidiabetic effects [1,12].

### 4.2. Carotenoids

Carotenoids are liposoluble pigments widely distributed in nature that can be found in plants, algae, fungi, and some bacteria. They have antioxidant and anti-inflammatory effects in addition to being precursors of vitamin A. They are present in nature in their trans form and through isomerization in the cis form, which occurs in an acid medium, in the presence of light, or at elevated temperatures. The isomerization of its molecule

can occur during processing, reducing its color and biological activity, as cis compounds have less potential for conversion into vitamin A due to the spatial conformation of this structure [21]. These characteristics make the extraction method a point of attention in the study so that compounds with important bioactive activities are not degraded during processing, and processes that use high temperatures, for example, are not indicated.

**Table 2.** Pulp and kernel fatty acids composition of *Acrocomia aculeata* [10,12,14].

| Components | Pulp | Kernel |
|---|---|---|
| Caprilic Acid (C8:0) % | | 6.28 |
| Capric Acid (C10:0) % | 0.15 | 4.78 |
| Lauric Acid (C12:0) % | - | 45.4 |
| Miristic Acid (C14:0) % | 0.22 | 9.5 |
| Palmític Acid (C16:0) % | 15.8 | 6.88 |
| Palmitoleic Acid (C16:1) % | 3 | - |
| Esterasic Acid (C18:0) % | 1.51 | 2.65 |
| Oleic Acid (C18:1) % | 71.76 | 24.05 |
| Linoleic Acid (C18:2) % | 1.79 | 3.02 |
| Linolenic Acid (C18:3) % | 1.01 | - |
| Cis-11-Eicosenic Acid (C20:1) % | 0.18 | 0.12 |
| Saturate Acids % | 18 | 75.63 |
| MCFA (cadeia média) % | 82 | 56.46 |
| MUFA (monoinsaturads) % | 74.94 | 24.17 |
| PUFA (polinsaturads) % | 2.80 | 3.02 |

The carotenoid profile in the pulp and seed of *Acrocomia aculeata* collected in different regions of Brazil showed about 11 carotenoids, with β-carotene (Figure 2) being the most abundant one with levels that varied significantly between regions from 106.69 to 265.23 μg/g in the pulp, while the seed presented similar results between regions with an average of 33.6 μg/g. Values higher than these were found in macaúba; authors identified a content of 694 μg/g of this carotenoid in macaúba [9]. The results indicate that macaúba is a potential source of carotenoids [1].

**Figure 2.** Molecular structure of β-carotene.

The concentration of carotenoids can vary according to the part of the fruit, degree of ripeness, and cultivation conditions. The carotenoid profile in macaúba was evaluated at three stages of ripeness, highlighting its importance as a promising nutritional source of lipophilic micronutrients. It was observed that the carotenoid profile of macaúba from Costa Rica differs significantly from the available data for samples from Brazil, suggesting genetic heterogeneity within the species [6,7]. The same study found 25 carotenoids in macaúba fruits at different stages of ripeness, including phytoene, phytofluene, zeaxanthin, anteraxanthin, and violaxanthin. The accumulation of carotenoid precursors and xanthophyll representatives cycle indicated the progress of the fruit ripening [7]. The study on the influence of the ripening stage on the content of β-carotene and lutein presence in the fruit indicates an increase in β-carotene and a decrease in lutein with the fruit ripening.

### 4.3. Phenolics Compounds

Phenolic compounds are substances that have a benzene nucleus and one or more hydroxyl groups. They can be divided into several classes (phenolic acids, flavonoids, tannins, lignins, and stilbenes). These compounds are known to have several biological activities,

including antioxidant, anti-inflammatory, antimicrobial, and anticancer effects [12,21], in addition to being secondary metabolites most commonly found in plants.

In *Acrocomia aculeata*, the most common compounds are tannins, flavonoids, catechins, steroids and/or triterpenoids, and saponins [11]. Among the phenolic products, caffeic acid, p-coumaric acid, and ferulic acid stand out, which are mainly present in the seeds and their oil. The flavonoids most often found in the species are catechins, quercetin, and luteolin, present in the leaves and fruits and responsible for the bitter taste of these parts of the plant [6].

Different authors found levels that varied between 1.4 and 2.69 mg GAEg$^{-1}$ oil (equivalent gallic acid) of total phenolic compounds in macaúba pulp oil [4].

### 4.4. Tocopherols

Recent studies highlight the presence of tocopherols and tocotrienols in its oils, known as forms of vitamin E. Tocotrienols are the main form of vitamin E in the seeds of the Arecaceae family. In the oils of *Acrocomia aculeata*, all tocotrienols and α-, β-, and δ-tocopherols were identified in the mesocarp [22]. The presence of tocopherols in significant amounts in the fruits of *Acrocomia aculeata* depends on the ripening stage, with α-tocopherol concentration progressively increasing during ripening [7,17]. Comparatively, the concentrations of α-tocopherol in *Acrocomia aculeata* fruits are notably higher than in other fruits such as apricots and avocados. Differences in vitamin E profiles in macaúba oils are attributed to genetic variation among wild populations and climatic conditions.

The pulp oil of macaúba can present a higher vitamin E content when grown in regions with higher precipitation rates, relative humidity, and temperature. This is an indication that this crop may be suitable for agroforestry systems in areas where it has not been considered due to climatic conditions [17,22]. The pulp oil of macaúba showed a tocopherol content of 212.95 mg/kg [4]. The most common form of vitamin E, α-tocopherol, was found in the analyzed fruits [17].

In summary, *Acrocomia aculeata* is a promising palm with potential for food, therapeutic, and industrial applications. Its oils are rich sources of vitamin E and carotenoids, with antioxidant and nutritional benefits. Variations in vitamin E profiles can be attributed to genetic variation and cultivation conditions [23].

### 5. *Acrocomia aculeata* Oil Extraction Using Supercritical Fluids

Supercritical fluid extraction is a technique used to obtain extracts with desired components from a solid or liquid material using a solvent under supercritical conditions, which occur above the critical point of the solvent where it exhibits intermediate properties between a liquid and a gas. When compared to other methods, supercritical fluid shows several advantages. In terms of economics, supercritical extraction can be easily scaled up to industrial levels with lower costs than other techniques. In terms of environmental impact, it is a green-friendly technique, where carbon dioxide is commonly used as a solvent due to its non-toxic, non-flammable, non-polluting, and highly selective properties, capable of extracting polar and nonpolar compounds by adjusting extraction conditions and adding co-solvents [15]. Thus, supercritical fluid extraction (SFE) technology allows selective extraction of specific compounds according to the pressure and temperature conditions used. In that regard, several studies have been developed with the aim of optimizing the efficiency of the process to obtain pure, concentrated, and high-quality extracts in short periods of time, an important factor for application in the pharmaceutical, chemical, and food industries [14,24].

Recent studies have shown that SFE with carbon dioxide is efficient in obtaining macaúba pulp and almond oils, providing greater selectivity and extraction of fatty acids, especially oleic acid, compared to low-pressure extractions such as Soxhlet [14]. On the other hand, extraction using compressed propane as a solvent presented promising results, with higher yields and higher levels of phytosterols and tocopherols in just 30 min of extraction [15].

*Oil Extraction Conditions from Acrocomia aculeata Pulp*

The extraction of *A. aculeata* oil from the fruit pulp has been investigated using different techniques and solvents, Table 3. In recent studies, supercritical carbon dioxide extraction was compared with conventional organic solvent extraction, such as n-hexane and ethanol. The results demonstrate that in SFE, operational conditions of temperature and pressure significantly affect the yield and composition of the obtained oil [14]. The authors verified that pressure variations at constant temperatures generate an increase in solvent density, which in turn increases the extraction yield due to the higher loads of solute in the solvent. However, at higher temperatures, the highest yield was found at lower pressures and, consequently, higher densities. This indicates that the increase in the solute vapor pressure had more influence on the increase in the extraction yield than the decrease in the solvent density. This phenomenon can be identified in the literature on several plant matrices [14]. Regarding Soxhlet and low-temperature extraction techniques, higher yields of macaúba oil occurred compared to SFC experiments. However, due to the high selectivity of $CO_2$, the supercritical extracts obtained for all experimental conditions showed higher levels of fatty acids, especially oleic acid (C(18:1)), compared to conventional extractions.

**Table 3.** Oil yields obtained (dry basis) in the extraction of macaúba pulp oil using different techniques and conditions.

| References | Methods | Solvents | Conditions | Solvent Density (kg/m$^3$) | Oil Yield (%) | β-Carotene (mg/100 g) | Flavonoids (mg/100 g) | Phytosterols (mg/100 g) |
|---|---|---|---|---|---|---|---|---|
| [25] | Soxhlet | Petroleum ether | 60 °C (480 min) | n.d. | 25.72 | 124.16 | 25.34 | - |
| | Low-pressure | Ethyl acetate n-hexane Isopropanol | 40 °C/40 rpm (240 min) | n.d. n.d. n.d. | 22.97 23.39 27.43 | 308.38 334.04 348.30 | 16.97 13.78 14.78 | 82.45 76.85 104.15 |
| [14] | Soxhlet | n-hexano Ethanol | 80 °C (360 min) 80 °C (360 min) | n.d. n.d. | 30.09 31.10 | - - | - - | - - |
| | Supercritical fluids (Nascimento, 2016) | Carbon dioxide | 40 °C/150 bar 40 °C/200 bar 55 °C/150 bar 55 °C/200 bar | 781.17 840.61 654.90 755.41 | 4.09 9.64 14.49 13.83 | - - - - | - - - - | - - - - |
| [15] | Soxhlet | n-hexano Dichloromethane | 70 °C (480 min) 40 °C (480 min) | n.d. n.d. | 25.64 26.83 | - - | - - | - - |
| | Supercritical fluids (Trentini, 2017) | Propane Propane Propane Propane Propane | 100 °C/40 bar 100 °C/120 bar 50 °C/40 bar 50 °C/120 bar 60 °C/80 bar | 110 410 440 470 420 | 9.77 22.69 22.86 23.08 22.8 | - - - - - | - - - - - | - - - - - |

Another study compared the oil yields in extraction under supercritical conditions using propane as a solvent and conventional extraction via Soxhlet using n-hexane and dichloromethane as solvents [15]. The authors show an increase in oil yield using propane compared to the other cited studies that used supercritical carbon dioxide as a solvent. In addition, the use of supercritical propane provided 86% of the yield obtained by conventional extraction, but the extraction time was only 30 min, while Soxhlet requires 480 min to reach approximately 26% yield.

It is important to note that in supercritical conditions, it is widely used as a solvent due to its cost, selectivity, and inertness, among other advantages. However, because it has low solubility to triacylglycerides, it requires high pressures and long extraction times to achieve a satisfactory yield. In this context, propane, which has a critical point under milder conditions (43 bar), has an advantage over carbon dioxide and may present higher oil yields in a shorter extraction time using a smaller volume of solvent [14,15,25]. Many authors point to advantages in the use of propane in relation to the extraction of phytosterols, tocopherols, and carotenoids when compared to supercritical carbon dioxide [26,27].

The literature is still scarce and presents many opportunities for studies involving supercritical technology for obtaining bioactive compounds and high-quality oil.

### 6. Biological Activity of *A. aculeata* Extract

*Acrocomia aculeata* is widely used by the local population due to its various medicinal and nutritional properties. In addition, it has been the subject of much scientific research in recent decades due to its biological activity attributed to the presence of bioactive compounds, such as fatty acids, carotenoids, flavonoids, and phenolic compounds [12]. These compounds are sensitive and susceptible to degradation in the presence of oxygen, acid, light, elevated temperatures, enzymes, metals, and reactive species [28].

*6.1. Antidiabetcs Efects*

The incidence of diabetes has a significant impact on society and healthcare systems, representing a high cost for society, as well as other chronic non-communicable diseases that require long-term treatment and care to prevent future complications. Studies suggest that this cost will increase by about 69% by 2030 [8]. Therefore, it is important to act preventively, identifying risk groups, as well as early diagnosis and treatment, as a priority.

In this preventive context of the disease aiming to reduce the economic burden caused by diabetes, the search for alternative sources of energy to replace carbohydrates is growing. Studies have evaluated the antidiabetic effects caused by the ingestion of macaúba almond oil by rats with type 2 mellitus diabetes, and the results showed a reduction in blood glucose levels (hypoglycemic effect) and in the deposition of fatty acids in the adipose tissue of the animals. *A. aculeata* oil is rich in medium-chain fatty acids that are easily catabolized, which can reduce body fat [10,29].

Clinical intervention tests with medium-chain fatty acids confirmed greater oxidation of fat because they are preferentially oxidized, generating body energy expenditure, which prevents their deposition in adipose tissue [10].

*6.2. Anti-Inflammatory and Diuretics Effects*

The oil from A. aculeata pulp (MPO) exhibited anti-inflammatory and antimutagenic activities, possibly related to its profile of fatty acids and antioxidant compounds, such as carotenoids [12]. Recent studies highlight the antioxidant and medicinal potential of Acrocomia aculeata, including its anti-inflammatory, diuretic, antidiabetic, and antimutagenic properties [14,30]. These properties may be related to the presence of bioactive compounds, such as fatty acids, phenolic compounds, flavonoids, and carotenoids. The exploitation of such properties may contribute to the development of new products based on natural products, such as alternative food supplements and treatments for diseases related to oxidative stress [17]. In addition, bocaiuva is rich in galactoglucomannan, a polysaccharide that also acts in anti-inflammatory processes [30].

Studies using microencapsulated oil from Acrocomia aculeata have shown significant effects of the fruit as a diuretic and anti-inflammatory in tests conducted on rats [30]. The authors evaluated the diuretic action of macaúba oil by analyzing diuresis (urine production) and the sodium content in the rats' urine. The results indicated a significant increase in urine production and sodium excretion, indicating a diuretic effect of *Acrocomia aculeata* oil.

*6.3. Antioxidant Effects*

Oxidative stress induced by free radicals and reactive oxygen/nitrogen species can lead to various health disorders such as DNA damage and inflammatory diseases. Therefore, a diet rich in antioxidants is important to inhibit oxidative stress. Therefore, macaúba presents a variety of medicinal and antioxidant properties that make it a valuable resource for health and well-being. Several studies have investigated the beneficial effects of this plant, especially regarding its antioxidant and medicinal properties [12].

Studies were carried out to determine the antioxidant activity in macaúba pulp oil by scavenging the free radical 2,2-diphenyl-1-picrylhydrazyl (DPPH) and ORAC (oxygen radical absorbance capacity), showing values of 23.89 µg mL$^{-1}$ and 42.02 µM TE/g fresh mass, respectively [4,31,32].

Macaúba pulp oil is a source of natural antioxidants such as β-carotene, which can bring health benefits. Its pulp is also rich in antioxidants such as β-carotene and α-tocopherol and monounsaturated fatty acids such as oleic acid, presenting anti-inflammatory properties and helping to reduce lipid peroxidation in mice [9]. Tocopherols have antioxidant activity, protecting unsaturated fatty acids from lipid oxidation and exerting the biological activity of vitamin E in the human body. In addition to their nutritional role as a precursor to vitamin A, carotenoids also have antioxidant effects, acting as a protective agent against cardiovascular diseases, certain types of cancer, neurological disorders, age-related macular degeneration, and cataracts. Studies demonstrate that β-carotene from macaúba pulp is highly bioavailable, further increasing bioavailability when consumed with oil [17]. They also suggest that the presence of phenolic and flavonoid compounds in the mesocarp of bocaiuva may contribute to its antioxidant activity.

### 6.4. Photoprotective Activity

Exposure to solar radiation is one of the main causes of the high rates of cancer worldwide. UV radiation generates reactive oxygen species, such as superoxide and hydrogen peroxide, which act on skin oxidation. Thus, research indicates that the antioxidant action of macaúba pulp oil promotes skin protection against lipid peroxidation [4,33].

The macaúba oil presents photoprotective activity and, therefore, can be applied to sunscreens. Studies have identified an increase in the sun protection factor (SPF) by 27 in sunscreens that used nanostructured carriers with macaúba kernel and pulp oil to prevent skin cancer and photoaging. The authors suggest that this protective effect against ultraviolet radiation is achieved thanks to the high levels of fatty acids, polyphenols, and β-carotene in *Acrocomia aculeata* [4].

### 6.5. Antimicrobial Activity

This section discusses studies that investigated the antimicrobial activity of macaúba leaf extracts. In more recent works, no antibiotic effect was identified for Gram-positive bacteria Staphylococcus aureus and Enterococcus faecalis, and Gram-negative bacteria Escherichia coli and *Pseudomonas aeruginosa*, as well as antifungal effects for *Candida albicans* and *Candida parapsilosis* [11,29]. On the contrary, another study that evaluated oil extracted from macaúba seeds in different strains of bacteria, including *Staphylococcus aureus* and *Escherichia coli*, showed positive results for antimicrobial activity. Additionally, the oil was effective against fungi of the *Candida genus* [34]. The authors suggest that macaúba oil may be used as a natural alternative for the treatment of infections caused by microorganisms, especially those that show resistance to conventional antibiotics.

However, it is important to note that further research is needed to determine the safety and efficacy of using macaúba oil in humans, as well as to better understand the mechanisms of antimicrobial action of this oil.

## 7. Technological Applications

Macaúba is a promising species for technological applications, Table 4, due to its nutritional and bioactive properties already explored in the previous sections. Thus, studies on technological applications of macaúba have focused mainly on the production of oil, which can be used in developing new products in the food, cosmetic, and pharmaceutical industries [12,13].

The species has sparked the interest of researchers and entrepreneurs due to its high productive potential and specific properties [35]. The oil extracted from the mesocarp of macaúba fruits contains carotenoids, such as β-carotene and β-cryptoxanthin, which are important for the nutraceutical and bioprocess industries [36]. However, the oil extraction process generates by-products, such as almond flour and defatted residual pulp flour, which also have significant nutritional and functional properties.

Macaúba almond flour is rich in carbohydrates, lipids, proteins, and ash. Sequilhos, a type of biscuit, were made with different proportions of macaúba almond flour and

showed satisfactory physicochemical and microbiological properties. In addition, macaúba residual flour can be incorporated into biscuit formulations, improving the nutritional quality of the final product [37,38]. The macaúba kernel isolate protein stands out for its high protein content (94.9%) and high levels of arginine (16.21%) and glutamate (20.84%). It also has nutritional properties and technological functionalities, such as oil and water retention capacity, as well as emulsification and gelation, which can be exploited by the food industry [39].

The residue from macaúba pulp oil extraction has also been evaluated as an alternative raw material for feeding growing pigs and can be included in pig diets without affecting the average daily gain or feed conversion, although it may result in animals with greater backfat thickness for higher inclusion rates [40].

Macaúba pulp and residual kernel flour have prebiotic potential, as they are rich in dietary fibers and antioxidants, which can benefit human health and contribute to the sustainability of food production [41]. The use of these residues in food production not only minimizes costs but also enriches the final products nutritionally [37].

Developing sustainable and knowledge-based value chains for macaúba requires a collaborative and systemic approach. To fully exploit macaúba's potential in the bio-economy, it is essential to develop a broad collaboration between scientists, public and private sectors, farmers, and civil society [35].

Macaúba has also been studied as a promising raw material for the production of sustainable alternative aviation fuels due to its high oil quality and good yield per hectare. The use of sustainable and less food-competitive raw materials is crucial for the development of sustainable aviation fuel chains [2]. Research has shown that these palms have significant potential to generate carbon credits and contribute to mitigating the effects of climate change [19]. Additionally, it has been shown that it is possible to produce biodiesel from macaúba seed oil using an eggshell catalyst, achieving an ester conversion yield of about 91% [41].

Other works have also explored the use of commercial acidic ion exchange resins in the synthesis of ethyl esters from crude macaúba oil, with promising results and good potential for biodiesel production using heterogeneous acid catalysts [42]. Moreover, various types of heterogeneous catalysts were evaluated for producing fatty acid methyl esters through interesterification reaction, with the mixed oxide of Ca-Mg-Al with 40% weight of Ca being the most suitable [43].

In summary, macaúba (*Acrocomia aculeata*) presents significant potential as a sustainable energy source and raw material in various areas, including food production and biofuels. The exploration of this neotropical palm species can contribute to the development of sustainable and diversified value chains, ensuring that sustainability and context aspects are considered at each stage of the process [35].

**Table 4.** Technology applications.

| Applications | References |
| --- | --- |
| Biodiesel | [44–46] |
| Nutraceutics foods | [36] |
| Foods (sequilhos-type biscuits, cookies, protein isolate) | [37,38,47] |
| Sustainable alternative fuels for aviation | [42] |
| Systems approach for the development of biomass-based value webs | [35] |
| Prebiotics | [41] |
| Animal foods | [40] |
| Cosmetics | [4] |
| Pharmaceuticals | [12,30] |
| Biolubricants | [2] |

## 8. Conclusions

The studies presented suggest that *Acrocomia aculeata*, or macaúba, has significant potential in a wide range of technological applications, from the food and pharmaceutical industry to the bioenergy sector.

Based on the reviewed studies, it is concluded that *Acrocomia aculeata*, known as macaúba, offers invaluable value as a promising source of carotenoids, particularly β-carotene and lutein. The data suggest that the carotenoid content in the plant may vary depending on the region of cultivation and the fruit's maturation stage, highlighting the relevance of a contextual approach in exploring its nutritional benefits. This, therefore, positions *A. aculeata* as a viable alternative for the production of carotenoid-rich foods.

Furthermore, the presence of antioxidant compounds such as flavonoids and phenolic acids contributes to the nutritional profile of macaúba and suggests significant potential for health benefits, such as preventing oxidative stress and reducing the risk of several chronic diseases. The investigation of these properties, however, requires additional studies to assess the therapeutic efficacy of these compounds in humans.

With regard to technological applications, macaúba demonstrates considerable potential, covering the production of oil, food, and even the production of biofuels. The supercritical extraction method for obtaining the oil stands out in this paper, which represents an area of opportunity for future research. This extraction method not only has the potential to optimize oil yield but also has the advantage of preserving the quality of the bioactive compounds present in macaúba oil, increasing its nutritional and functional value. Despite the promise, supercritical extraction requires additional studies to determine the ideal process conditions.

Ultimately, the sustainable exploitation of macaúba has the potential to contribute to building diverse and sustainable value chains. However, for its potential to be fully harnessed, a collaborative effort involving scientists, the public and private sectors, farmers, and civil society is needed, ensuring that sustainability is incorporated into all stages of the process.

**Author Contributions:** Conceptualization, writing—original draft preparation and supervision, G.C.M.A. and R.N.d.C.J.; formal analysis and writing—review and editing, G.C.M.A. All authors have read and agreed to the published version of the manuscript.

**Funding:** This study was financed in part by the Coordenação de Aperfeiçoamento de Pessoal de Nível Superior—Brasil (CAPES)—Finance Code 001 and by the Pró-Reitoria de Pesquisa e Pós-Graduação PROPESP/UFPA (PAPQ).

**Institutional Review Board Statement:** Not applicable.

**Informed Consent Statement:** Not applicable.

**Data Availability Statement:** Data are contained within the article.

**Acknowledgments:** The authors acknowledge the Federal University of Pará (UFPA through PROPESP, Brazil) and Coordenação de Aperfeiçoamento de Pessoal de Nível Superior—Brazil (CAPES). The authors acknowledge that Maria Eduarda Ferraz de Carvalho, who was responsible for translating this article.

**Conflicts of Interest:** The authors declare they have no conflict of interest.

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
