# Peer review of "Potential of Supercritical Acrocomia aculeata Oil and Its Technology Trends"

_applsci, doi:10.3390/app13158594_

Round 1

Reviewer 1 Report

PFA

Author Response

Response to Reviewer 1 Comments

“After careful review, I have concluded that your article requires major revision before it can be considered for publication.”

Response: We appreciate the positive feedback and we are thankful for your kind and stimulating words.

 Point 1: “Authors need to provide a more compelling title”

Response: Thanks for the suggestion. The title of the work was changed: “Potencial of Supercritical Acrocomia aculeata oil and its technologies trends” (lines 2 and 3). In fact, the new title is more attractive and consistent with the data presented in the article.

Point 2: “Authors must provide more information subject of the review especially on the phytochemical profile and biological effects and others.”

Response: Thank you for your suggestion, it is very pertinent. Macaúba (Acrocomia aculeata) is an extremely abundant species in the Amazon region, its potential is well known and widespread for application in the preparation of biodiesel. However, this fact shows great research opportunities to be carried out with macaúba as a source of oil for the pharmaceutical, cosmetic and food industries. We added a table (Table 3 in lines 267 - 269) with results on bioactive compounds evaluated in macaúba extracts, in addition we inserted a paragraph in lines 299 - 301 that brings information that we consider important to be presented to the reader.

Point 3: “This article appears to be a mini review rather than a systemic review”

Response: Thanks for the suggestion. Evaluating carefully, in fact the article is a mini review. In that regard, we made the change in the category of the article.

Reviewer 2 Report

The authors reviewed Phytochemical profile, fatty acid composition, biological effects and potential for supercritical extraction of Acrocomia aculeata: A Review. I can recommend the paper for a possible publication.  This paper aims to provide a comprehensive evaluation of Acrocomia aculeata.

Good work. But I have following observations.

Check the abstract for grammatical errors, and it needs modification.

The language is to be checked throughout the paper.

Improve the abstract

Please include subtitle regarding the supercritical extraction process and parameters of Acrocomia aculeata.And Please write this part which parameters use for extraction and include advantages and disadvantages

There is no link within the conclusion. It needs to be modified.

A proper representation of tables is needed.

Please check the journal format and revise Manuscrit accroding to journal format

The language is to be checked throughout the paper.

Main text should be grammar improve and in some cases it is very weak and maybe there is no discussion at all.

Author Response

Response to Reviewer 2 Comments

“The authors reviewed Phytochemical profile, fatty acid composition, biological effects and potential for supercritical extraction of Acrocomia aculeata: A Review. I can recommend the paper for a possible publication.  This paper aims to provide a comprehensive evaluation of Acrocomia aculeata.

Good work. But I have following observations”.

Response: We appreciate the positive feedback and we are thankful for your kind and stimulating words.

Point 1: Check the abstract for grammatical errors, and it needs modification.

Response: Thanks for the observation, we made the necessary adjustments in the abstract for better understanding of the reader.

Point 2: The language is to be checked throughout the paper.

Response: Thank you for your note. All adjustments were made throughout the text.

Point 3: Improve the abstract

Response: Thanks for the suggestion. All the adjustments that we deem necessary for the correct understanding of the article by the reader have been made (lines 11 – 24).

Point 4: Please include subtitle regarding the supercritical extraction process and parameters of Acrocomia aculeata. And Please write this part which parameters use for extraction and include advantages and disadvantages

Response: Thanks for the suggestion. We made a change in the title of section 5 which deals with the extraction of oil from Acrocomia aculeata by supercritical fluid (line 207). Additionally, we made modifications to the text and added in this section a sub-item with information that demonstrates the results obtained in several studies of oil extraction from macaúba pulp (lines 232, 235), indicating the different extraction conditions, methodology and advantages of these processes (lines 239-249). In addition, we added two more paragraphs that show the effects on the obtained oil yield results and the advantages of replacing carbon dioxide with propane in supercritical extraction (lines 250-265), in addition to two necessary references (lines 568-573) to the basis of the text.

Point 5: There is no link within the conclusion. It needs to be modified.

Response: Thanks for the suggestion. We made changes to completion lines 443 – 469. In this way, it established a greater contribution to the academic community.

Point 6: A proper representation of tables is needed.

Response: Thanks for the suggestion. We made the changes we deemed necessary in lines 120, 235, 383 and 437. In addition to the necessary inclusion of one more table (table 3 in lines 267-269) in the section “Oil extraction conditions from Acrocomia aculeata pulp”.

 Point 7: Please check the journal format and revise Manuscrit accroding to journal format.

Response: Thanks for the suggestion. The article was revised based on the journal format.

Round 2

Reviewer 1 Report

Accept